# Left ventricular longitudinal strain variations assessed by speckle-tracking echocardiography after a passive leg raising maneuver in patients with acute circulatory failure to predict fluid responsiveness: A prospective, observational study

**Clemence Roy**[1], **Gary Duclos**[1], **Cyril Nafati**[2], **Mickael Gardette**[2], **Alexandre Lopez**[1], **Bruno Pastene**[1], **Eliott Gaudray**[1], **Alain Boussuges**[3], **François Antonini**[1], **Marc Leone**[1], **Laurent Zieleskiewicz**[1,3]*

1 Department of Anesthesiology and Intensive Care Unit, Aix-Marseille University, Assistance Publique Hôpitaux de Marseille, Hôpital Nord, Marseille, France, 2 Department of Anesthesia and Intensive Care Medicine, University Hospital of Marseille, la Timone Hospital, Marseille, France, 3 Center for Cardiovascular and Nutrition Research (C2VN), INSERM, INRA, Aix Marseille Université, Marseille, France

* laurent.zieleskiewicz@ap-hm.fr

## Abstract

### Background

An association was reported between the left ventricular longitudinal strain (LV-LS) and pre-load. LV-LS reflects the left cardiac function curve as it is the ratio of shortening over dia-stolic dimension. The aim of this study was to determine the sensitivity and specificity of LV-LS variations after a passive leg raising (PLR) maneuver to predict fluid responsiveness in intensive care unit (ICU) patients with acute circulatory failure (ACF).

### Methods

Patients with ACF were prospectively included. Preload-dependency was defined as a velocity time integral (VTI) variation greater than 10% between baseline (T0) and PLR (T1), distinguishing the preload-dependent (PLD+) group and the preload-independent (PLD-) group. A 7-cycles, 4-chamber echocardiography loop was registered at T0 and T1, and strain analysis was performed off-line by a blind clinician. A general linear model for repeated measures was used to compare the LV-LS variation (T0 to T1) between the two groups.

### Results

From June 2018 to August 2019, 60 patients (PLD+ = 33, PLD- = 27) were consecutively enrolled. The VTI variations after PLR were +21% (±8) in the PLD+ group and -1% (±7) in the PLD- group (*p*<0.01). Mean baseline LV-LS was -11.3% (±4.2) in the PLD+ group and

**Data Availability Statement:** The data file is held in the Open Science Framework repository at https://osf.io/e6tm4/.

**Funding:** The authors received no specific funding for this work.

**Competing interests:** Laurent Zieleskiewicz and Marc Leone, co-authors, declared competing interests that do not alter our adherence to PLOS ONE policies on sharing data and materials.

-13.0% (±4.2) in the PLD- group ($p$ = 0.12). LV-LS increased in the whole population after PLR +16.0% (±4.0) ($p$ = 0.04). The LV-LS variations after PLR were +19.0% (±31) ($p$ = 0.05) in the PLD+ group and +11.0% (±38) ($p$ = 0.25) in the PLD- group, with no significant difference between the two groups ($p$ = 0.08). The area under the curve for the LV-LS variations between T0 and T1 was 0.63 [0.48–0.77].

## Conclusion

Our study confirms that LV-LS is load-dependent; however, the variations in LV-LS after PLR is not a discriminating criterion to predict fluid responsiveness of ICU patients with ACF in this cohort.

## Introduction

Acute circulatory failure (ACF) is a major issue occurring in up to 65% of intensive care unit (ICU) patients [1]. ACF can result from several mechanisms (hypovolemic, cardiogenic, distributive shocks) requiring different interventions. Fluid resuscitation (FR) is the first-line treatment for patients with preload-dependency and is used to expand the intravascular compartment and thus improve cardiac output (CO) and end-organ perfusion [2, 3]. Clinical evaluation alone leads to inappropriate FR in up to 50% of cases [4, 5]. Yet, inappropriate FR has been associated with increased morbidity (i.e. acute respiratory failure, acute kidney injury, abdominal compartment syndrome) and mortality [6–10]. Therefore, FR should be guided by repeated assessments of patients' hemodynamic status [2] using markers of preload-dependency to predict fluid responsiveness. Many preload-dependency markers have been studied recently, and those based on dynamic change of cardiac output or surrogates are considered as standard of care to evaluate preload-dependency in patients with ACF [11].

Among the existing preload-dependency markers, a variation greater than 10% in left ventricular outflow track velocity-time integral (LVOT-VTI) measured with transthoracic echocardiography (TTE) during a passive leg raising (PLR) maneuver is considered a reliable method [11–15]. This marker is non-invasive, easily accessible at patients' bedside in ICUs. However, the evaluation of the LVOT-VTI variations requires cautious interpretation due to its inter-individual variability [16–18].

Longitudinal speckle-tracking strain echocardiography is widely used by cardiologists, especially in patients with heart failure [19, 20] and is increasingly used in ICUs [21–24]. This technique is based on the tracking of acoustic markers (called speckle) within the myocardium during the cardiac cycle. This technology allows for an assessment of the myocardium strain during systole and diastole [25] and is a highly reproducible procedure, as shown by Negishi *et al* [26]. The longitudinal strain (LS) value for each myocardial segment is defined by the following formula: $LS = (L-L_0)/L_0$, where L is length of the segment during systole, and $L_0$ is the length of the segment in end-diastole. The longitudinal strain has a negative value, with a normal range for the left ventricle (LV) of around -16% to -19% [27].

The aim of this technology is first to describe the systolic segmental function of the myocardium [28] but modified load conditions were found to be associated with modified LV-LS value, especially when changes in preload occur [21, 29–33]. Nafati *et al*. evaluated the LV-LS in preload-dependent (PLD+) ICU patients and found a decreased mean LV global longitudinal strain (LV-GLS) value of -13.3% [21]. In this study, LV-GLS values normalized after FR, confirming that LV-GLS depends on preload conditions. Indeed, the longitudinal strain

formula refers to myocardial fibers' deformation. We attempted to assess the hypothesis suggesting that, when a PLD+ patient undergoes PLR, the result is an increased LV preload, and thus, an extension of "$L_0$" (increased LV volume in end-diastole). A shortening of "L" during systole then ensues, by improvement of LV inotropism according to the Franck-Starling principle. Thus, PLD+ patients should have detectable LV longitudinal strain variations in TTE after PLR.

The aim of this study was thus to determine the sensitivity and specificity of LV-LS variations after PLR to predict fluid responsiveness in ICU patients with ACF, using LVOT-VTI variations after PLR as a reference method to assess preload dependency.

## Methods

This study was prospective and observational and took place in a 15-beds ICU at North Hospital in Marseille, France from June 2018 to August 2019. The protocol was approved by the Ethics Committee of Amiens, France (IRB 2017-A03584-49, May 4, 2018). Informed written consent was obtained from either the patients or their relatives, after an oral and written information was delivered by the clinician, according to the French law [34]. Patients admitted to the ICU with ACF upon admission or during the ICU stay were screened. The inclusion criteria included any of the following clinical or biological features: arterial hypotension (systolic blood pressure < 90 mmHg or mean blood pressure (MAP) < 65 mmHg), urine output < 0.5 ml/kg/h, use of vasopressors to keep MAP > 65 mmHg, clinical signs of hypoperfusion (*i.e.* mottled skin) and serum lactate concentration > 2.0 mmol/l. Patients were either under mechanical ventilation, non-invasive ventilation or spontaneous breathing. The exclusion criteria included: age < 18 years, patient under guardianship, patient already included in another trial, arrythmia or non-sinus cardiac rhythm, mitral regurgitation, right ventricle (RV) heart failure, elevated left atrial pressure, cardiogenic pulmonary edema, restrictions on passive leg raising realization (unstable rachis trauma, intra-cranial hypertension, pericardial tamponade, aortic dissection), lack of echogenicity or bandages covering the TTE area, pregnancy, patient's refusal to participate.

In patients eligible for inclusion, TTE was performed to assess preload-dependency, a standard care procedure for patients with ACF at our ICU. The features (age, sex, weight, height, reason for admission), cardiovascular co-morbidities, sepsis related organ failure assessment (SOFA) score, simplified acute physiology score (SAPS) II, and serum lactate concentrations were recorded for each patient after enrollment. Systolic, diastolic and mean arterial pressures and heart rate were recorded at each time point. The inclusion time was the duration of the echocardiographic assessment at the patient's bedside (approximatively 10 min). Each patient had an inclusion number, used to register loops and echocardiographic features anonymously in the ultrasound machine. Patients were screened and included successively, whenever an echocardiography-board certified clinician (CR, GD, LZ) was available to enroll the patient and perform TTEs according to the protocol described below.

### Time-point schedule

First TTE (T0) was performed in patients at a semi-recumbent position of 45˚ (Fig 1). Echocardiographic measurements with LVOT-VTI assessments and 4-chambers loops were recorded. Then, a PLR was performed, and the clinician recorded a second LVTO-VTI (T1) within 60 seconds after the beginning of PLR [13]. The patients were classified in the "preload-dependent" group (PLD+) if the LVOT-VTI variations after PLR exceeded 10% [12]. Otherwise, they were classified into the "preload-independent" group (PLD-).

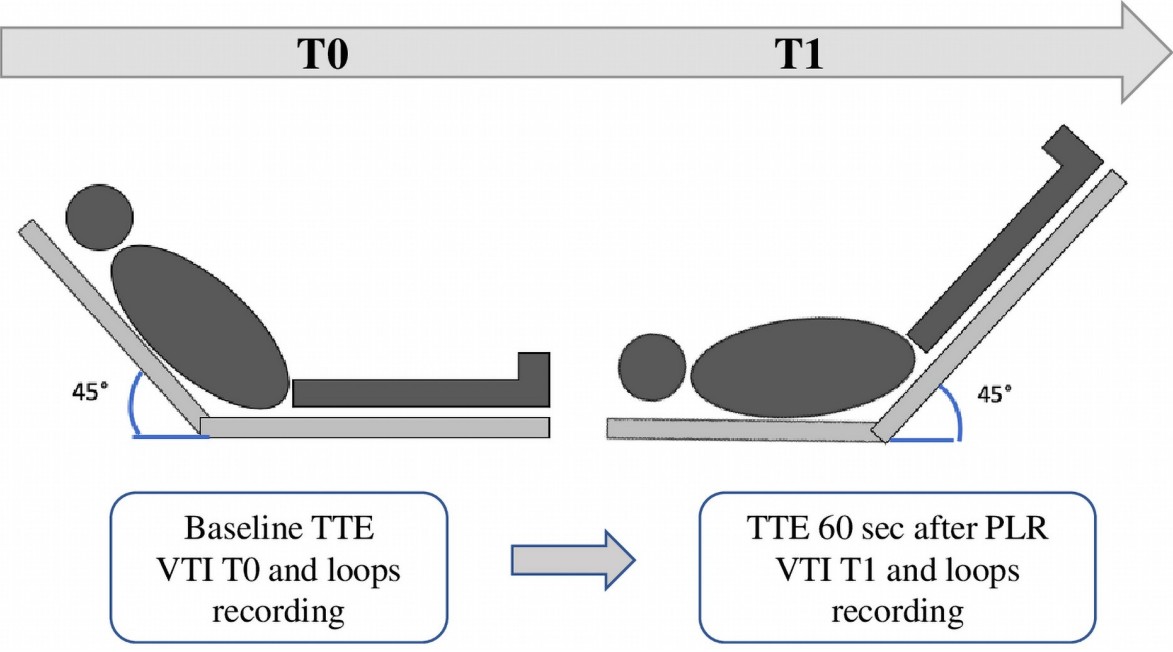

**Fig 1. Time-point schedule and echocardiography protocol.** TTE: trans-thoracic echocardiography; VTI: velocity time integral; PLR: passive leg raising.

## Echocardiography protocol

TTEs were performed by echocardiography-board certified clinicians (CR, GD, LZ) [35] using a General Electric (GE) Vivid IQ machine (GE HealthCare, Chicago, Illinois, USA) and a 3.5-Hz probe. The conventional evaluation was performed following the American Society of Echocardiography Recommendations [36]. A cross-sectional area of the aortic anulus was calculated from its diameter during early systole, measured in a parasternal long-axis view. The other variables were collected using an apical 4 or 5-chamber view. The ejection fraction was estimated visually by the operators [37] who also checked if the RV/LV ratio was < 1 and right ventricle had an homogeneous and normal kinetics. Echogenicity was judged as poor, moderate or good according to the ability of the clinician to either see LV with no access to RV, see LV and access to RV on another view or have a perfect visualization of the four myocardial cavities within the same view.

At each time point (T0 and T1), LVOT-VTI was measured using pulse-wave Doppler from an apical 5-chamber view. Each reported measurement of LVOT-VTI was an average of three to five consecutive measurements over one respiratory cycle. Mitral flow was assessed using an apical 4-chamber view with pulse-wave Doppler allowing for the measurement of the *E* and *A* wave velocities and the *E/A* ratio.

Using tissue Doppler imaging via an apical 4-chamber view, the velocity of the lateral mitral annulus *E'* wave and the *E/E'* lateral ratio were calculated. At the end of each time-point, a 4-chamber 7-cycles loop was registered in DICOM format with a frame rate above 50/second, for further longitudinal strain assessment. Considering the transient effect of PLR to eventually modify load conditions during T1, the operator rapidly stored loops and doppler images, and made the measurements afterwards. The whole echocardiography evaluation was assessed by the same operator for each patient. The strain analysis was conducted off-line on a decentralized computer, via the EchoPac™ clinical workstation software version 202 (GE Healthcare,

Chicago, Illinois, USA) by level 2 (CR) and level 3 (LZ) operators [38] trained in 2D-strain echocardiography. Each anonymized TTE loop was analyzed by an operator blinded to the patient's group (PLD+ or PLD-).

The operator traced the LV myocardial contour using the semi-automatized method of speckle tracking after identification of the baso-septal, baso-lateral and apical points [39] and adjusted the contouring manually if necessary after the visualization of the tracking on dynamic loops. LV-LS was calculated three times and averaged for each time-point loop. Intra-observer reproducibility was calculated from these data. Fifteen loops were randomly selected for a double LV-LS analyze by two clinicians (CR, LZ) to calculate the inter-observer reproducibility. To make our analysis relevant for clinical practice and due to the good correlation between the global longitudinal strain (GLS) value calculated from the averaged 2, 3 and 4-chamber views or from a 4-chamber view alone [21], longitudinal strain (LS) was estimated from the six LV segments of the apical 4-chamber loop only. The systolic strain rate (SSR) was also calculated for each loop. Its value was the most negative value of the strain rate curve occurring after the opening of the aortic valve. We also assessed right ventricle longitudinal strain (RV-LS) in the 4-chamber loops allowing for a good visualization of the entire myocardial wall. We performed an averaged measurement of strain values for the three segments of the RV lateral wall obtained through the 4-chamber view. Left atrial (LA) longitudinal strain was also analyzed following a standardized method when the registered loops allowed a good visualization of LA [40].

## Statistical analysis

Data analysis was performed using R-Project for Statistical Computing 2.14 (The R Foundation, Vienna, Austria). Categorical variables were expressed as numbers and percentages (%) and continuous variables were expressed as mean ± standard deviation (SD). For each patient, VTI variations or strain variations were defined by *(Value at T1 –Value at T0) / Value at T0*. Statistical analysis consisted of a univariate and a bivariate analysis comparing the PLD+ and PLD- groups. Continuous variables were compared using a Kruskal-Wallis test. A *p* value $< 0.05$ was considered significant. A general linear model for repeated measures was used to compare LV-LS variations (between T0 and T1) between the two groups. In this model, PLD status (+ or -) was the predictor and LV-LS was the outcome. We assumed these LV-LS variations would be a valuable clinical tool if the area under the curve (AUC) was above 0.85 with a 95% confidence interval (CI) from 0.75 to 0.95. For this purpose, 60 patients had to be included.

The intra-observer reproducibility of LV-LS measurements was calculated from LV-LS values at T0 measured 3 times on the same loops by the same operator, on 15 patients. The mean difference was calculated and divided by the mean of the three values. The inter-observer reproducibility was calculated after LV-LS was measured three times and averaged on the same T0 loop in 15 patients by two clinicians (CR, LZ). Again, the mean difference was calculated and divided by the mean of the two clinicians mean value. Standard deviation was obtained from the three consecutive measurements of LV-LS on T0 loop for each patient, and the corresponding coefficient of variation (CV) was calculated as CV = SD/mean of the three measurements.

## Results and discussion

During the study period, 109 patients met the inclusion criteria but 49/109 (44%) also met the exclusion criteria, notably poor echogenicity or no TTE window (n = 17, 35%), arrythmia

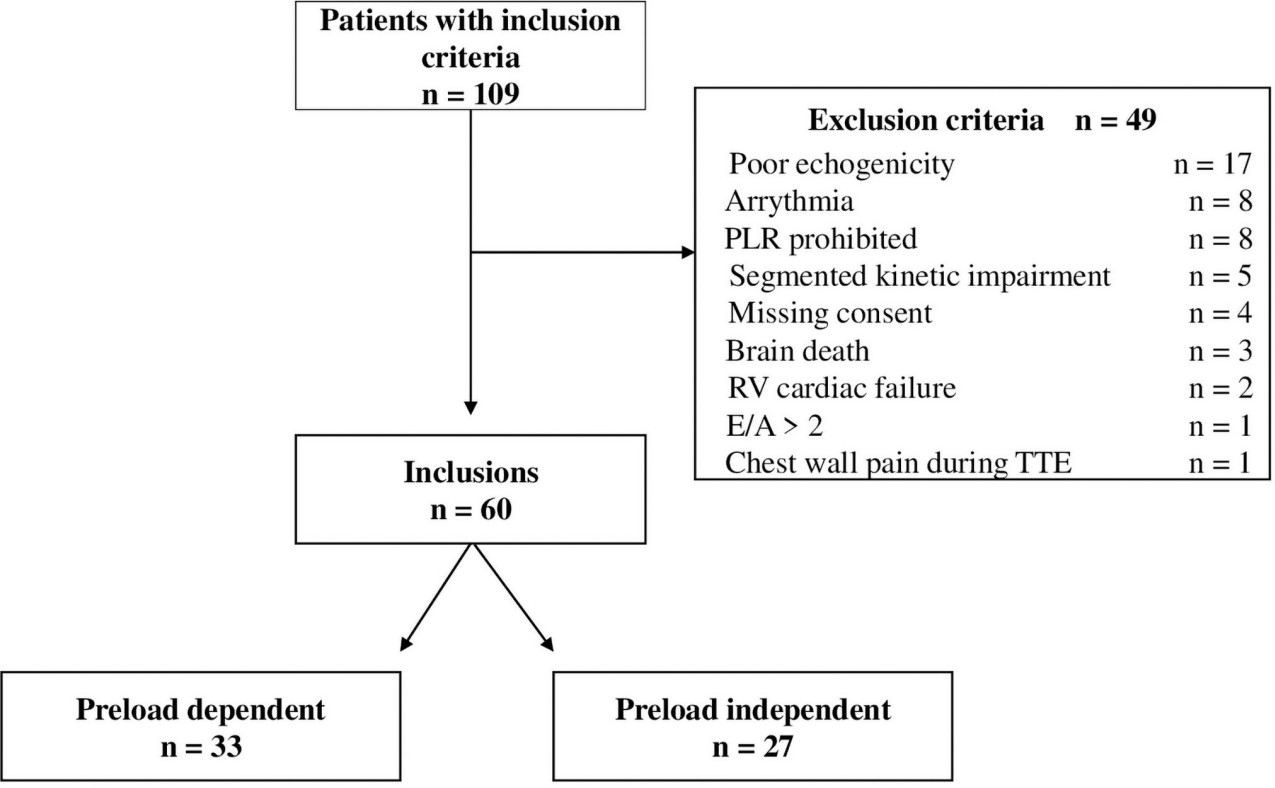

**Fig 2. Flow chart.** PLR: passive leg raising maneuver, RV: right ventricle, LV: left ventricle, TTE: trans-thoracic echocardiography.

(n = 8, 16%) and prohibited PLR (n = 8, 16%). From June 2018 to August 2019, 60 patients, 33 in the PLD+ group and 27 in the PLD- group, were prospectively included (Fig 2).

## Characteristics of patients

At baseline, the patients' characteristics were similar for all features except for vasopressors administration (more frequent in the PLD- group) and shock aetiology. Septic shock was found for 10 (30%) patients in the PLD+ group and 16 (59%) patients in the PLD- group ($p = 0.03$). In contrast, hemorrhagic shock was reported in 10 (30%) patients in the PLD+ group versus 2 (7%) in the PLD- group ($p = 0.04$) (Table 1).

The clinical and biological variables were similar, apart from the initial heart rate, which was higher in the PLD+ group than in the PLD- group (Table 2).

## Echocardiographic data

At T0, the LVOT-VTI values were lower in the PLD+ group than in the PLD- group (16.8 cm (± 5.0) versus 20.6 cm (± 4.2), $p< 0.01$) whereas heart rate was significantly higher in the PLD+ group than in the PLD- group, resulting in a similar cardiac output. The other echocardiographic data at T0 were similar between the two groups. Mean baseline LV-LS was -11.3% (± 4.2) in the PLD+ group and -13.0% (± 4.2) in the PLD- group ($p = 0.12$) (Table 2).

Regarding variations after PLR, mean LVOT-VTI variations (T0-T1) were +21% (±8) in the PLD+ group and -1% (±7) in the PLD- group ($p< 0.01$). Mean LV-LS variations between T0 and T1 in the whole population were +16% (±4) ($p = 0.04$). In the PLD+ group, mean LV-LS variations between T0 and T1 were +19% (±31) ($p = 0.05$) whereas in the PLD- group,

**Table 1. Patient characteristics at baseline.**

|  | All n = 60 | PLD+ (n = 33) | PLD- (n = 27) | *p* |
|---|---|---|---|---|
| Sex F/M | 25/35 | 10/23 | 15/12 | 0.08 |
| Age (years) | 58.1 ± 16.8 | 54.7 ± 19.7 | 62.2 ± 11.4 | 0.09 |
| Mechanical ventilation [*n* (%)] | 34 (56%) | 17 (51%) | 17 (63%) | 1 |
| PEEP (mmHg) | 6.0 ± 2.0 | 6.1 ± 2.2 | 6.2 ± 2.2 | 0.94 |
| ARDS [*n* (%)] | 13 (22%) | 6 (18%) | 7 (26%) | 0.68 |
| RASS score | -2 ± 2 | -2 ± 3 | -2 ± 2 | 0.85 |
| Vasopressor infusion [*n* (%)] | 44 (73%) | 20 (60%) | 24 (80%) | 0.03 |
| Norepinephrine infusion (mg/h) | 1.0 ± 1.5 | 1.0 ± 1.8 | 1.1 ± 1.2 | 0.07 |
| Lactate at t0 (mmol/l) | 2.7 ± 2.5 | 2.7 ± 2.7 | 2.7 ± 2.2 | 0.74 |
| SOFA score | 7.8 ± 3.6 | 7.9 ±3.9 | 7.8 ±3.2 | 0.98 |
| SAPS II score | 47.8 ± 18.6 | 49.8 ± 17.9 | 45.3 ±19.4 | 0.37 |
| Cardiovascular comorbidities [*n* (%)] |  |  |  |  |
| Arterial hypertension | 15 (25%) | 7 (21%) | 8 (29%) | 0.65 |
| Coronary artery disease | 3 (5.0%) | 2 (6%) | 1 (3%) | 1 |
| Valvular disease | 5 (6.7%) | 3 (9%) | 1 (3%) | 0.75 |
| Stroke | 2 (3.3%) | 0 | 2 (7%) | 0.38 |
| Shock aetiology [*n* (%)] |  |  |  |  |
| Sepsis | 26 (45%) | 10 (30%) | 16 (59%) | 0.03 |
| Vasoplegia without sepsis | 7 (12%) | 1 (3%) | 6 (22%) | 0.02 |
| Acute haemorrhage | 12 (20%) | 10 (30%) | 2 (7%) | 0.04 |
| Hypovolemia | 14 (23%) | 11 (33%) | 3 (11%) | 0.06 |
| Cardiogenic | 1 (1.7%) | 1 (3%) | 0 | 1 |
| Multiple trauma patient [*n* (%)] | 21 (35%) | 16 (48%) | 5 (18%) | 0.03 |
| Associated organ failure [*n* (%)] |  |  |  |  |
| Neurological | 32 (53%) | 19 (57%) | 13 (48%) | 0.64 |
| Respiratory | 32 (53%) | 15 (45%) | 17 (63%) | 0.27 |
| Kidney | 12 (20%) | 5 (15%) | 7 (26%) | 0.47 |
| Coagulation | 14 (23%) | 7 (21%) | 7 (26%) | 0.90 |
| Liver | 6 (10%) | 2 (6%) | 4 (7%) | 0.49 |

Data are expressed as numbers and rate or as mean ± SD. PLD+: preload-dependent, PLD-: preload-independent, PEEP: positive end-expiratory pressure, ARDS: acute respiratory distress syndrome, RASS: Richmond Agitation-Sedation scale, SOFA: Sepsis-related Organ Failure Assessment, SAPSII: simplified acute physiology score.

LV-LS variations were +11% (±38) (*p* = 0.25). These variations of LV-LS after PLR were not statistically different between the groups (*p* = 0.08) (Fig 3).

Using a linear model for repeated measures, LV-LS variations after PLR were not different between the two groups (*p* = 0.13) (Fig 4). There was no significant difference neither after adjusting the model on two possible confounding factors, septic shock and norepinephrine infusion.

The AUC for the LV-LS variations between T0 and T1 was 0.63 [0.48–0.77], which suggests that the variations in LV-LS after PLR do not predict fluid responsiveness in patients with ACF (Fig 5).

Right ventricular longitudinal strain was assessed in 27 patients (16 from the PLD+ group and 11 from the PLD- group) showing no difference between baseline and T1 in the whole population (RV-LS at T0 was -14.6 (±5.38) and at T1–14.7 (±6.55), *p* = 0.94). There was no difference between the two groups at T0: mean RV-LS was -14.7% (±5.8) in the PLD+ group

**Table 2. Clinical and echocardiographic data before (T0) and during (T1) the passive leg raising maneuver.**

| | PLD+ (t0) n = 33 | PLD- (t0) n = 27 | *p* (t0) | PLD+ (t1) n = 33 | PLD- (t1) n = 27 | *p* (t1) |
|---|---|---|---|---|---|---|
| Echogenicity | | | 0.88 | | | |
| Poor (n =) | 3 | 3 | | | | |
| Moderate (n =) | 15 | 17 | | | | |
| Good (n =) | 9 | 13 | | | | |
| Heart rate (b/min) | 97 ± 21 | 83 ± 18 | <0.01 | 95 ± 21 | 82 ± 17 | <0.01 |
| SAP (mmHg) | 100 ± 19 | 108 ± 18 | 0.16 | 110 ± 22 | 112 ± 21 | 0.60 |
| DAP (mmHg) | 54 ± 13 | 54 ± 11 | 0.87 | 58 ± 12 | 56 ± 11 | 0.52 |
| MAP (mmHg) | 67 ± 12 | 71 ± 12 | 0.19 | 74 ± 14 | 72 ± 19 | 0.87 |
| LVOT diam (mm) | 20 ± 2.0 | 19 ± 2.0 | 0.14 | | | |
| LVOT VTI (cm) | 16.8 ± 5.0 | 20.6 ± 4.2 | <0.01 | 20.1 ± 6.0 | 20.4 ± 4.0 | 0.35 |
| CO (L/min) | 5.0 ± 1.7 | 4.8 ± 1.2 | 0.85 | 5.9 ± 1.8 | 4.9 ± 1.3 | 0.03 |
| CI (L/min/m$^2$) | 2.8 ± 1.0 | 2.7 ± 0.7 | 0.85 | 3.3 ± 1.1 | 2.8 ± 0.7 | 0.11 |
| LVEF (%) | 59 ± 11 | 54 ± 11 | 0.08 | | | |
| S lateral (cm/s) | 11.7 ± 3.9 | 10.4 ± 2.7 | 0.34 | 11.1 ± 3.3 | 9.8 ± 2.3 | 0.23 |
| E' lat (m/s) | 0.1 ± 0.0 | 0.1 ± 0.0 | 0.86 | 0.1 ± 0.0 | 0.1 ± 0.1 | 0.70 |
| E/A | 1.0 ± 0.3 | 1.0 ± 0.3 | 0.67 | 1.0 ± 0.4 | 1.0 ± 0.3 | 0.99 |
| E/E' lat | 7.4 ± 2.6 | 7.7 ± 2.9 | 0.79 | 7.0 ± 2.3 | 7.3 ± 3 | 0.99 |
| IVC Variation (%) | 26 ± 2.0 | 30 ± 2.0 | 0.53 | | | |
| LV–LS (%) | -11.3 ± 4.2 | -13.0 ± 4.2 | 0.12 | -13.3 ± 5.2 | -13.8 ± 4.1 | 0.51 |
| LV–SSR (s$^{-1}$) | -0.9 ± 3.0 | -0.9 ± 4.0 | 0.67 | -1.0 ± 0.4 | -0.9 ± 0.3 | 0.65 |
| RV–LS (%) | -14.7 ± 5.8 | -14.4 ± 5.0 | 0.80 | -16.4 ± 6.6 | -12.3 ± 5.8 | 0.09 |
| LA–LS (%) | 18.2 ± 7.0 | 15.0 ± 6.0 | 0.23 | 16.7 ± 6.6 | 14.4 ± 5.6 | 0.28 |

Data are expressed as mean ± SD. p(t0) is p value for comparison of means between PLD+ and PLD- group at T0, p(t1) is p value for comparison of means between PLD + and PLD- group at T1. SAP: systolic arterial pressure, DAP: diastolic arterial pressure, MAP: mean arterial pressure, LVOT: left ventricle outflow track, VTI: velocity time integral, CO: cardiac output, CI: cardiac index, LVEF: left ventricle ejection fraction, S lateral: peak systolic wave in tissue doppler at the lateral mitral annulus, E' lat: peak early diastolic lateral mitral annulus velocity, E: peak early diastolic transmittal flow velocity, E/A ratio of E to A, E/E'lat ratio of E to E′lat, IVC: inferior vena cava, LV-LS: left ventricular longitudinal strain, LV-SSR: left ventricular systolic strain rate, RV-LS: right ventricular longitudinal strain, LA-LS: left atrial longitudinal strain.

and -14.4% (±5.0) in the PLD- group (*p* = 0.80). The RV-LS variations after PLR were +10% (±45%) in the PLD+ group and +5% (±61%) in the PLD- group (*p* = 0.69).

At baseline, mean SSR in the whole population (n = 60) was -0.90 (±0.36) and did not vary after PLR; -0.96 (±0.35), *p* = 0.37. SSR was similar in the 2 groups (PLD+ -0.90s$^{-1}$ (±3); PLD- −0.90s$^{-1}$ (±4); *p* = 0.67) and did not vary after PLR in either of the groups.

Baseline mean left-atrial longitudinal strain (LA-LS) was 16.69 (±6.51) in the whole population (available evaluation for 31 patients) and did not vary after PLR; 15.64 (±6.17), *p* = 0.41. Neither did this variable vary after PLR between the two groups.

Intra-observer variability was 6.6% (±4.6) (CR level 2 operator) and 8.3% (±6.2) (LZ level 3 operator). Coefficient of variation was 5.3% (±3.7) for CR and was 5.2% (±5.1) for LZ. Inter-observer variability (calculated with means of three measurements performed by each observer) was 4.4% (±3.1) and coefficient of variation was 2.2% (±1.6).

## Discussion

To our knowledge, this is the first study evaluating the performance of LV-LS variations to predict fluid responsiveness in ICU patients. Our results confirm previous findings showing that LV-LS is preload-dependent, as LV-LS value increased significantly in the whole population

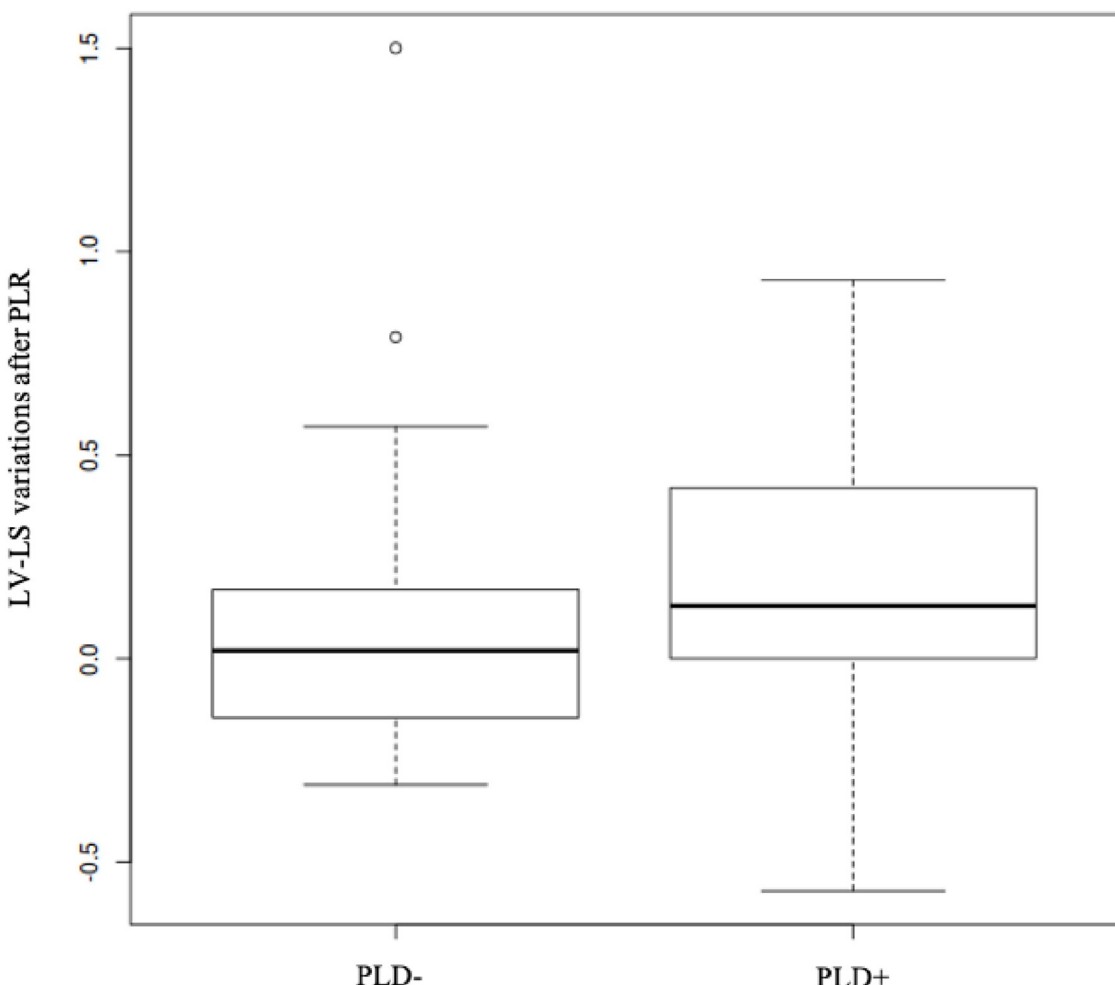

**Fig 3. Variations of mean LV-LS after a passive leg raising.** + 11% (±38%) in the PLD- group versus + 19% (±31%) in the PLD+, *p* = 0.08 (Kruskal-Wallis test). Dark line represents the median for each group.

and in the PLD+ subgroup after a PLR maneuver. However, in our cohort, LV-LS variations during PLR were not a discriminating marker to assess preload-dependency and predict fluid responsiveness in clinical practice.

In their study, Nafati *et al*. showed that absolute LV-LS values at baseline were altered in a preload-dependent population of ICU patients [21]. Our study confirms these results, with a decreased baseline LV-LS in our entire population. Mean LV-LS was worst in the PLD+ group but increased in both groups after PLR (+19% in the PLD+ group *p* = 0.05; +11% in the PLD- group *p* = 0.25), which made it difficult to ascertain a significant difference in LV-LS augmentation between the two groups. These LV-LS variations appeared to be mainly related to the increase in the end-diastolic stretch of the myocardial fibers ($L_0$) after correcting preload via PLR, as shown previously, but this stretching response can also depend on the aetiology of ACF.

The studied population in this cohort is representative of patients admitted to ICUs with ACF, with varied pathologies leading to circulatory failure such as sepsis, haemorrhage, major surgery responsible for a systemic inflammatory response. We chose to enroll patients admitted for any cause of ACF, which led to a great heterogenicity in our population, regarding

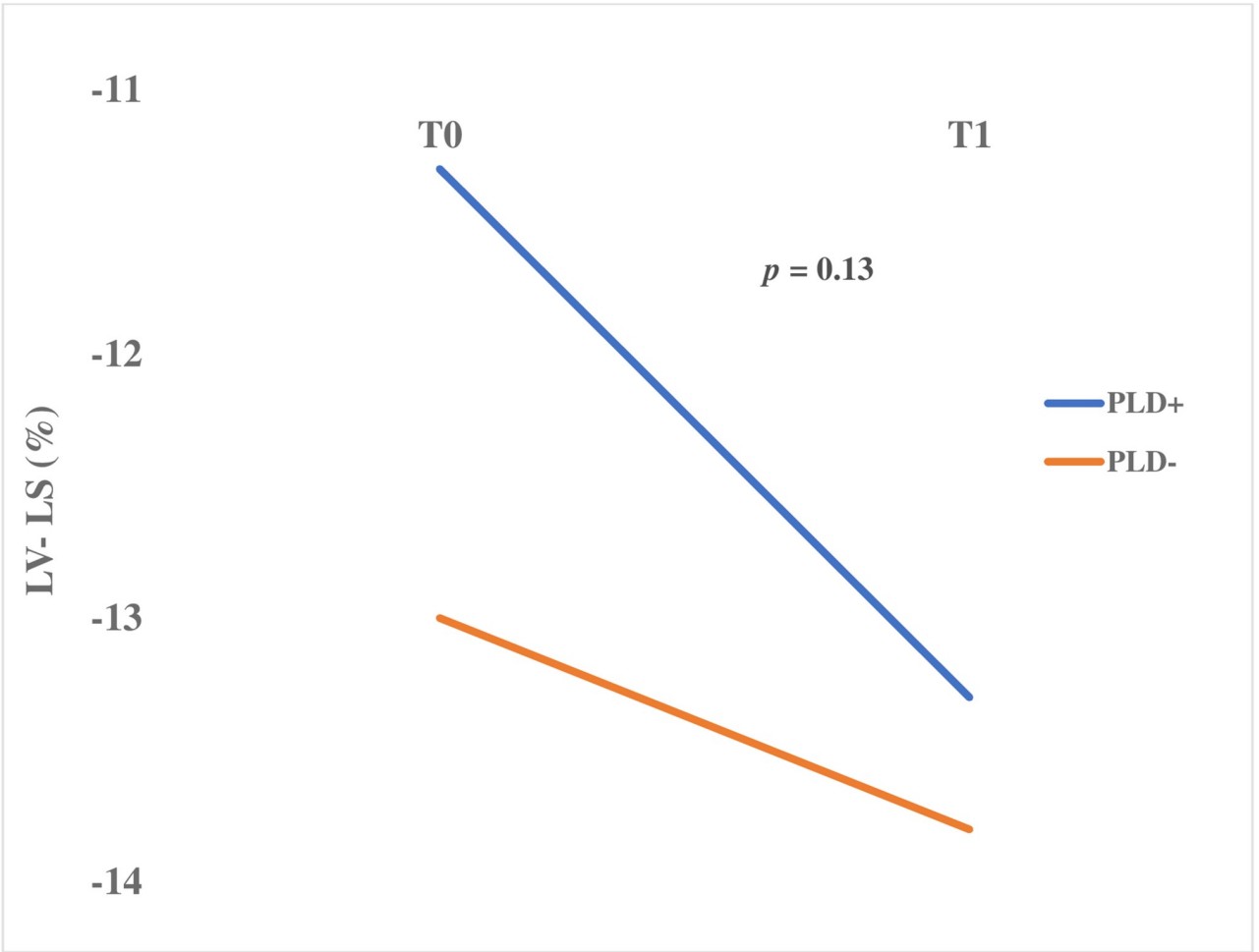

**Fig 4. Compared LV-LS variations in-between baseline and PLR, in PLD+ patients versus PLD- patients, using a linear model for repeated measures, *p* = 0.13.**

clinical features and the pathophysiological mechanism responsible for a circulatory failure. Our final groups, PLD+ and PLD-, were not akin regarding the aetiology of ACF.

As a matter of fact, there were significantly more patients admitted for septic shock in the PLD- than in the PLD+ group, which possibly biased our analysis. Boissier *et al.* showed that patients with septic shock could be either hyperkinetic with improved LV-LS values or normo-kinetic or hypokinetic with lower LV-LS values [41]. They explained these results through the various incidences of myocardial dysfunction in septic shock patients and the heterogeneity of afterload conditions, but they assessed LV-LS without an accurate evaluation of preload (i.e. through LV filling pressure, vena cava variation, end-diastolic and systolic volume estimation). Ng *et al.* compared a group of patients with septic shock with a control group of septic patients without shock [42]. Again, there were significant differences in LV-LS, but no accurate evalua-tion of preload-dependency was performed. To date, it has not been possible to assert that septic shock is responsible for an alteration in LV-LS that is independent of load conditions. Nevertheless, septic shock induces changes in both preload and afterload, which interact with inotropism. The resulting LV dysfunction is typically associated with non-elevated filling pres-sures and increased LV compliance [43, 44]. These pathophysiological changes in LV function

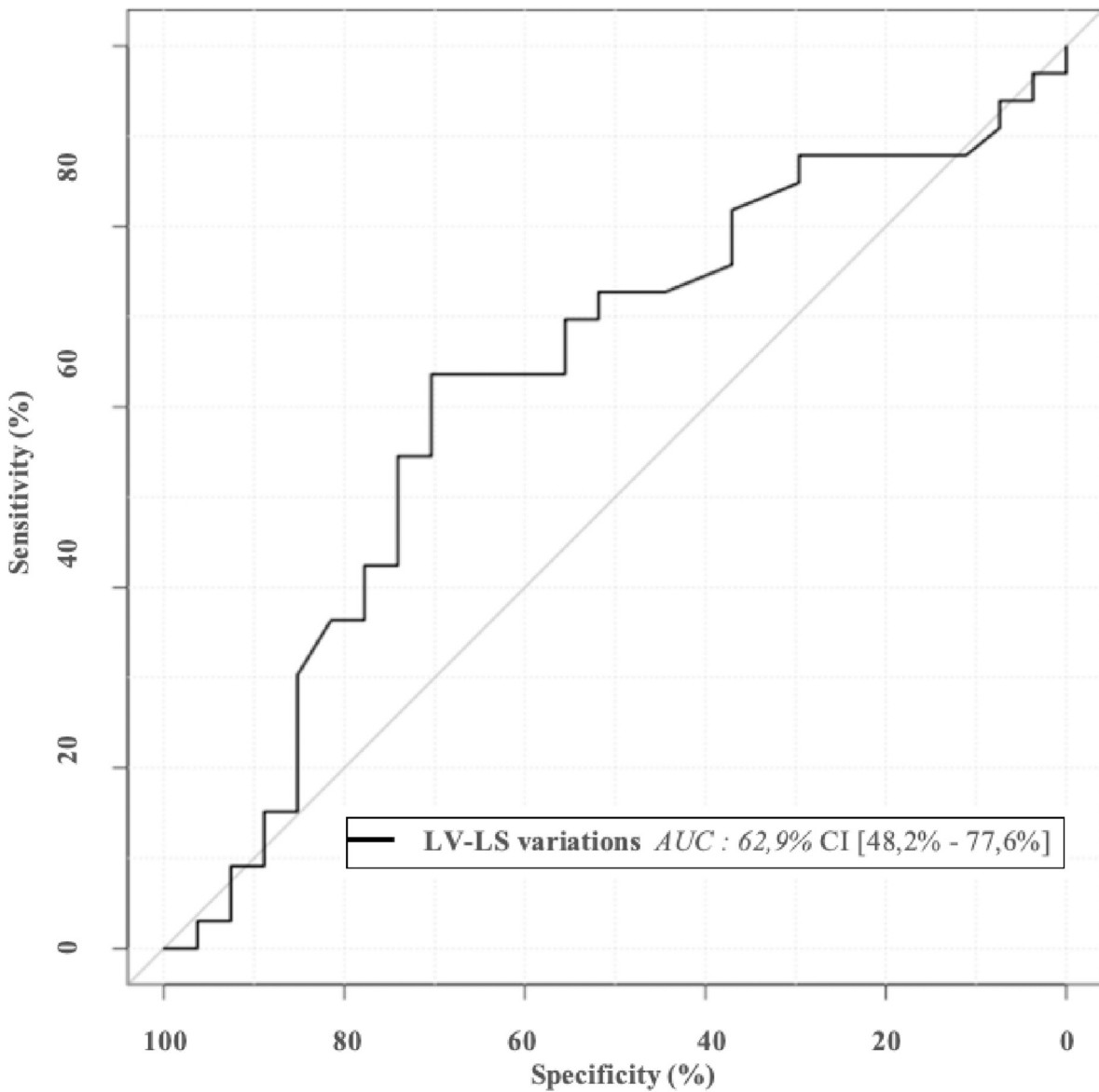

**Fig 5. Receiver operating curve for evaluation of LV-LS variations during PLR to predict fluid responsiveness.** AUC: area under curve.

in septic patients affect LV-LS. In our study, the septic population was too small to conduct relevant subgroup analysis, but this repartition of septic patients could have influenced our findings. Moreover, statistical analysis revealed a wide spread of data, making it difficult to conclude to significant differences and to define a threshold for clinical use.

We must also acknowledge a lack of statistical power to ascertain a correlation between LV-LS variations and preload even in the "hypovolemia" subgroup, due to a small number of patients. We basically calculated the number of required subjects according to our primary goal, which was to determine the sensitivity and specificity of LV-LS variations to assess pre-load-dependency with an area under the curve of at least 0.85. We could not anticipate how LV-LS would vary in our cohort and in what proportion. The *a posteriori* calculated required number of subject to ascertain a statistical difference in LV-LS variation between the groups explains a lack of power in our study.

Data for RV-LS was only available for 27 patients (16 PLD+ and 11 PLD-) at both T0 and T1. Regarding the RV, our results suggest a tendency for an increased RV-LS in the PLD + group whereas RV-LS decreased in the PLD- group after PLR. Our findings are limited due to a low number of patients; however, they do foster the launch of future studies.

Systolic strain rate (SSR) was also measured at each time-point. We found no differences in SSR between the two groups and no variations after PLR. Fredholm *et al.* found that SSR depends on both preload and heart rate and found an increase after PLR. However, they performed strain measurements via transesophageal echocardiography during cardiac surgery with cardiopulmonary bypass, which involves very different loading conditions than those met in our ICU patients [45].

Regarding the usual echocardiographic markers that were evaluated at T0 and T1, there was a significant difference concerning cardiac output (and surrogates as LVOT-VTI) between the PLD+ and PLD-, but no significant difference was observed concerning echocardiographic markers of left-ventricular end diastolic pressure (LVEDP). Indeed, there was no difference between mean E, E' lat, nor E/E'lat between PLD+ and PLD- and no significant variation of these markers between T0 and T1 in the whole population. Yet, some authors showed a relationship between E'lat and preload. Mahjoub *et al.* [46] showed a significant increase of E'lat (and a resultant decrease in E/E'lat) in a population of septic shock patients with diastolic dysfunction (E'Lat < 0.12cm/s) receiving a 500 ml crystalloid administration. These results suggest an improvement of LV relaxation with the correction of hypovolemia in patients with septic shock and diastolic dysfunction. Lamia *et al.* [14] studied the variations of echocardiographic markers of preload and LVEDP using a PLR maneuver, followed by a volume expansion and found similar results as ours, i.e. a non-significant variation of E, E'Lat and E/E'lat between responders and non-responders following PLR or crystalloid infusion. This study also concerned patients with various aetiology responsible for circulatory failure. Thus, if an increase of E'lat can be observed in preload-dependent patients after correcting hypovolemia, E/E'lat variation is not very accurate to quantify small transitory variations of preload and should rather be used as a marker of LVED, to anticipate the risk of fluid overload when a volume expansion is considered to treat ACF.

Our study has some other limitations we must acknowledge. Left-ventricular longitudinal strain was assessed using a 4-chamber echocardiographic view alone. The accurate assessment of GLS involves performing a 2-chamber, 3-chamber and 4-chamber view to average the values of all 17 LV myocardial segments. In clinical practice, this reference method is probably too complicated to assess fluid responsiveness. Furthermore, a recent study showed that LV global longitudinal strain calculated from averaged 2-3-4-chamber views had a strong correlation with LV longitudinal strain calculated from a 4-chamber view alone [21]. By extension, we considered LV-LS the LV longitudinal strain we obtained from a 4-chamber view.

We chose VTI variations after PLR as a reference method, with a 10% threshold to classify our patients into either the PDL or in the PLD- group. This threshold was acknowledged to accurately predict fluid responsiveness in ACF patients in most situations [12], including spontaneous breathing, atrial fibrillation and pregnancy [5, 47]. Yet, other authors like Roger *et al.* have used a higher threshold of 15% VTI variation to predict fluid responsiveness [48]. For that purpose, they performed a fluid infusion with 500 ml of crystalloids for every patient with ACF. They showed that preload-dependency is a labile parameter, with patients qualified as transient responders to FR 10 minutes after FR, becoming non-responders only 20 minutes after the end of FR. In our study, we used a non-invasive method to assess fluid responsiveness (*i.e* PLR), instead of administrating fluid on clinical criteria of ACF with a 50% risk of inappropriate FR. Our lower threshold might have misclassified patients into the PLD+ group.

Monnet *et al.* reviewed the performance of PLR in predicting fluid responsiveness in a meta-analysis. They showed that direct evaluation of CO variations (or surrogate like VTI variations) was the most accurate way to assess preload-dependency [12]. However, the evaluation of preload-dependency with PLR could be inaccurate if intra-abdominal pressure is above 12 cmH$_2$O. In our study, intra-abdominal pressure was not measured before inclusion, which could involve a possible classification bias in our patients [49]. Moreover, a study showed that, even when performed by the same operator, the least significant changes in the VTI assessments were 11%, suggesting the use of higher thresholds for VTI changes to predict fluid responsiveness [47].

## Conclusion

Our study showed that LV-LS variations after PLR do not predict fluid responsiveness.

Our results confirm that LV-LS is preload-dependent, but its value might also be influenced by afterload and intrinsic myocardial function, which varies with the aetiology of ACF. The heterogeneity of ACF causes in our cohort made it tenuous to draw conclusions about using speckle tracking strain variations as a useful tool to predict fluid-responsiveness or tolerance to fluid administration. An assessment of the relationship between LV-LS variations and preload in a more homogenous population of ICU patients might thus be of interest and requires further evaluation.

## Author Contributions

**Conceptualization:** Clemence Roy, Cyril Nafati, Mickael Gardette, Alexandre Lopez, Alain Boussuges, Marc Leone, Laurent Zieleskiewicz.

**Data curation:** Clemence Roy, Gary Duclos, Cyril Nafati, Eliott Gaudray, Laurent Zieleskiewicz.

**Formal analysis:** Clemence Roy, François Antonini.

**Methodology:** Clemence Roy, Cyril Nafati, Eliott Gaudray, François Antonini, Marc Leone, Laurent Zieleskiewicz.

**Project administration:** Clemence Roy.

**Supervision:** Marc Leone, Laurent Zieleskiewicz.

**Validation:** Gary Duclos, François Antonini, Marc Leone, Laurent Zieleskiewicz.

**Writing – original draft:** Clemence Roy, Laurent Zieleskiewicz.

**Writing – review & editing:** Clemence Roy, Gary Duclos, Cyril Nafati, Mickael Gardette, Alexandre Lopez, Bruno Pastene, François Antonini, Marc Leone, Laurent Zieleskiewicz.

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
