## [Decision Letter · Decision Letter 0]

24 May 2021

PONE-D-21-11995

Left ventricular longitudinal strain variations assessed by speckle-tracking echocardiography after a passive leg raising maneuver to predict fluid responsiveness: A prospective, observational, monocentric study

PLOS ONE

Dear Dr. ROY,

Thank you for submitting your manuscript to PLOS ONE. After careful consideration, we feel that it has merit but does not fully meet PLOS ONE’s publication criteria as it currently stands. Therefore, we invite you to submit a revised version of the manuscript that addresses the points raised during the review process.

This is an interesting paper that I think makes an important contribution to the literature in this area. However, there are some areas that I believe require revision and/or clarification prior to publication. In addition to the comments from the reviewers, please consider the following:

-Can you provide more detail about the repeated measures regression model? It seems like LV-LS was the outcome and PLR + vs - was the predictor--is this correct? While many of the characteristics were similar in the PLR groups, I wonder if there are any potential confounders that could be adjusted for in the model? there is probably not a "right" answer, and an overfit model is certainly not advantageous to anyone, but I think that more detail about the model itself and the decisions and assumptions that went into to would provide clarity and strength to the paper.

-Another important idea is that of patient-level heterogeneity and its relationship to PLR status and echo parameters. you have alluded to this in the conclusion section, and i certainly agree with the ideas presented therein. however, i think this is a very important part of the paper and would be more appropriately located in the discussion or limitations section, perhaps with a bit more detail/explanation.

We look forward to receiving your revised manuscript.

Kind regards,

Robert Ehrman, MD, MS

Academic Editor

PLOS ONE

Journal Requirements:

2. Please provide additional details regarding participant consent.

In the ethics statement in the Methods and online submission information, please ensure that you have specified what type of consent you obtained (for instance, written or verbal, and if verbal, how it was documented and witnessed). If the need for consent was waived by the ethics committee, please include this information.

'Laurent Zieleskiewicz received fees from General Electrics Healthcare for ultrasound teaching'

a. Please confirm that this does not alter your adherence to all PLOS ONE policies on sharing data and materials, by including the following statement: "This does not alter our adherence to  PLOS ONE policies on sharing data and materials.” (as detailed online in our guide for authors http://journals.plos.org/plosone/s/competing-interests).  If there are restrictions on sharing of data and/or materials, please state these.

Please note that we cannot proceed with consideration of your article until this information has been declared.

Reviewers' comments:

Reviewer's Responses to Questions

**Comments to the Author**

1. Is the manuscript technically sound, and do the data support the conclusions?

Reviewer #1: Partly

Reviewer #2: Yes

2. Has the statistical analysis been performed appropriately and rigorously? 

Reviewer #1: Yes

Reviewer #2: Yes

3. Have the authors made all data underlying the findings in their manuscript fully available?

Reviewer #1: Yes

Reviewer #2: Yes

4. Is the manuscript presented in an intelligible fashion and written in standard English?

Reviewer #1: Yes

Reviewer #2: Yes

5. Review Comments to the Author

Reviewer #1: I thank the authors and editor for the privilege of reviewing this manuscript. In, “Left ventricular longitudinal strain variations assessed by speckle-tracking echocardiography after a passive leg raising maneuver to predict fluid responsiveness: A prospective, observational, monocentric study,” Roy and colleagues assessed left ventricular longitudinal strain as a predictor for fluid responsiveness after a passive leg raise.

Major:

1. The PLR maneuver can cause translation of the heart and affect VTI assessments. Please comment on this in the limitations

2. The study omitted one third of eligible patients due to technical limitations. This may bias the sample, as these patients might be sicker. Please comment on this in the limitations.

3. I have concerns for a type I error. E/e’ is a surrogate for preload, or more specifically LVEDP. There is no difference between PLD+ and PLD- patients at T0 for E/e’. E/e’ is a marker of preload, not preload responsiveness, so it’s not completely incorrect if there is no difference between groups. However, E/e’ decreased with the PLR in both PLD+ and PLD- patients, which would suggest inconsistency in E/e’ measurement, in PLD assessment, or both. Please clarify. Additionally, the main outcome (ΔStrain after PLR) is perhaps not the most obvious variable to select, and it barely achieves significance, while several other hypotheses were assessed. Please confirm in the methods that the delta longitudinal strain after a PLR was the a priori primary assessment, or please correct these tests of significance for multiple hypothesis testing.

Minor:

1. Page 4 and 5. VTI after PLR is perhaps the most generalizable and generally the most accurate method of assessing preload responsiveness, but it is misleading to call it the gold standard. Please revise.

2. Table 1: Sex is misspelled (Sexe)

Reviewer #2: 1.This study included patients with ACF and echocardiography was conducted at T0 and T1. In conclusion, this study confirms that LV-LS is load-dependent; however, the variations in LV-LS after PLR is not a discriminating criterion to predict fluid responsiveness of ICU patients with ACF.

2.Title: It would be better to point out the subjects (patients with ACF) included in this study in the title. Additionally, the title should indicated the conclusion that variations in LV-LS after PLR is not a discriminating criterion to predict fluid responsiveness.

3.Methods: fluid responsiveness should be defined in the Methods.

4.Results(page13, line271): This study compared the variation of LV-LS in whole subjets after PLR. The variations of other parameters after PLR such as LV-SSR, RV-LS and LA-LS which were assessed by speckle tracking analysis should be added in this analysis to test their prediction value for fluid responsiveness.

5.Results(page13, line270-274): Significant differences of mean LV-LS variations between T0 and T1 was observed in the whole population (n=60, p=0.04), but not in the PLD+ group (n=33, p=0.04) and PLD- group(n=27, p=0.25). Additionally, these variations of LV-LS after PLR were not statistically different between the groups (p=0.08). The results might result from the small numbers of subjects, and statistical power test should ba added in this study.

6.Is there any liner correlation between varations of LV RVOT VTI and LV-LS after PLR?

7.Conclusion(page18, line384-385): This study did not test the correlations between LV-LS and afterload and intrinsic myocardial function. You should add related statistical analysis or modify the conclusion.

8.What is the advantages and innovations of this study?

6. PLOS authors have the option to publish the peer review history of their article (what does this mean?). If published, this will include your full peer review and any attached files.

Reviewer #1: No

Reviewer #2: No

---

## [Author Response · Author response to Decision Letter 0]

6 Jul 2021

* Comments from academic editor : 

1) Can you provide more detail about the repeated measures regression model? It seems like LV-LS was the outcome and PLR + vs - was the predictor--is this correct? 

While many of the characteristics were similar in the PLR groups, I wonder if there are any potential confounders that could be adjusted for in the model? there is probably not a "right" answer, and an overfit model is certainly not advantageous to anyone, but I think that more detail about the model itself and the decisions and assumptions that went into to would provide clarity and strength to the paper Thank you for your remark. In the presented regression model for repeated measures, preload dependency status (represented by PLD+ or PLD-) was the predictor, and the myocardial strain response to PLR, evaluated by LV-LS, was the outcome.

Response : In this model, LV-LS is the dependent variable, that varies between two time points T0 and T1. PLD+ and PLD- are independent of T0 and T1. The intra-subject effect shows an increase of LV-LS absolute value, in both groups (PLD+ and PLD-) between T0 and T1 (with p = 0.001 in the regression model). But there is no inter-subject statistical difference, with p = 0.13 when we compared the two groups.

We conducted a univariate analysis which found no association between the major confounding factors (e.g aetiology of ACF, norepinephrine infusion, lactate level, sedation). The correlation matrix showed no evidence for correlation, apart for expected couples (i.e septic shock/norepinephrine infusion).

We conducted further statistical analysis as you suggested, adjusting the regression model on two variables which were close to significance in the univariate analysis with p<0.2 : septic shock and norepinephrine infusion. 

When adjusting the regression model for repeated measures on septic shock we found p = 0.292, when adjusting on norepinephrine infusion we find p = 0.612.

We discussed this important point in the revised manuscript. 

2) Another important idea is that of patient-level heterogeneity and its relationship to PLR status and echo parameters. You have alluded to this in the conclusion section, and I certainly agree with the ideas presented therein. however, I think this is a very important part of the paper and would be more appropriately located in the discussion or limitations section, perhaps with a bit more detail/explanation 

Response : Thank you for this comment.

Changes were made in the discussion section.

“The studied population in this cohort is representative of patients admitted in ICUs with ACF, with varied pathologies leading to circulatory failure such as sepsis, haemorrhage, heavy surgery followed by systemic inflammatory response. We chose to enroll patients admitted for any cause of ACF, which led to a great heterogenicity in our population, regarding clinical features and the pathophysiological mechanism responsible for a circulatory failure. Our final groups, PLD+ and PLD-, were not akin concerning the aetiology of ACF. As matter of fact, there were significantly more patients admitted for septic shock in the PLD- than in the PLD+ group, which possibly biased our analysis.” [15]

*Comments from Reviewer #1

1) The PLR maneuver can cause translation of the heart and affect VTI assessments. Please comment on this in the limitations 

Response : Thank you for this remark we for sure agree with. 

However, it has been shown in a meta-analysis by Cherpanath et al. [15] that the pulse contour derived-cardiac output, as well as the cardiac output measured by esophageal Doppler, transthoracic echocardiography or bioreactance had a similar diagnostic performance to assess cardiac output variations after a passive leg raising maneuver. 

Possible translation of the heart during PLR does not seem to have a negative impact on the evaluation of CO surrogates such as LVOT-VTI. Furthermore, in clinical practice, delta VTI after PLR was acknowledged a validated tool with an almost perfect area under the curve (Monnet et al. [11]) and is currently consider as a gold standard for fluid responsiveness diagnosis in large studies (Vignon et al, Comparison of Echocardiographic Indices Used to Predict Fluid Responsiveness in Ventilated Patients, AJRCCM 2017). 

2) The study omitted one third of eligible patients due to technical limitations. This may bias the sample, as these patients might be sicker. Please comment on this in the limitations. 

Response : Thank you for this interesting question. 

Within the 49 excluded patients, 17 had poor echogenicity. Most of them were admitted after major thoracic surgery which is an important part of our ICU recruitment. Bandage, surgical wounds, leaft us poor echographic window. 

However, excluded patients had a similar mean SAPSII score to included patients. Unfortunately, we did not assess SOFA score in the non-included population but severity of the patients was never the reason of their non-inclusion. 

3) I have concerns for a type I error. E/e’ is a surrogate for preload, or more specifically LVEDP. There is no difference between PLD+ and PLD- patients at T0 for E/e’. E/e’ is a marker of preload, not preload responsiveness, so it’s not completely incorrect if there is no difference between groups. However, E/e’ decreased with the PLR in both PLD+ and PLD- patients, which would suggest inconsistency in E/e’ measurement, in PLD assessment, or both. Please clarify. Additionally, the main outcome (ΔStrain after PLR) is perhaps not the most obvious variable to select, and it barely achieves significance, while several other hypotheses were assessed. Please confirm in the methods that the delta longitudinal strain after a PLR was the a priori primary assessment, or please correct these tests of significance for multiple hypothesis testing. Thank you for this comment.

Response : As far as E/e’ is concerned:

We agree that E/e’ variation after PLR can be used to control PLR efficiency. However, E/e’ variation is not very accurate to quantify small transitory variations of preload. One reason might be that an increase of e’ is observed in preload dependent patients after correcting hypovolemia as shown by Mahjoub and colleagues (Mahjoub et al, Improvement of left ventricular relaxation as assessed by tissue Doppler imaging in fluid-responsive critically ill septic patients, Int Care Med, 2012).

As reviewer#1 noticed, E/e’ was more specifically validated as a marker of LV filling pressure, allowing clinicians to anticipate the risk of fluid overload when a volume expansion is considered to treat ACF.

Studies focused on evaluation of preload dependency found no association between E/e’ and fluid responsiveness. For example, Lamia et al ([14], Table3) found a similar E/e’ ratio between responder and non-responder patients, before and after volume expansion. Thus, we are not surprised to find a non-significant decrease of this parameter, after PLR, in both our groups. 

Second part of Rewiewer1 remark:

We confirm our main outcome was ΔLV-Strain after PLR, and the study was designed on the hypothesis that ΔStrain after PLR would be greater in PLD+ group than in PLD-. Our hypothesis was already raised in the paper from Nafati and colleagues [21] showing that LV-GLS value increased after volume expansion in preload dependent patients. 

Statistical analysis were based and designed on this only hypothesis. 

4) Page 4 and 5. VTI after PLR is perhaps the most generalizable and generally the most accurate method of assessing preload responsiveness, but it is misleading to call it the gold standard. Please revise.

Response : Thank you for this comment.

Because the area under curve of VTI variations after PLR is almost perfect to predict fluid responsiveness, we chose this marker as a Gold standard to avoid unnecessary fluid loading in preload independent patients. However, according to your comment, we removed this term in the all manuscript.

5) Table 1: Sex is misspelled (Sexe). 

Response : Noted.

*Comments from Reviewer#2

1) Title: It would be better to point out the subjects (patients with ACF) included in this study in the title. Additionally, the title should indicated the conclusion that variations in LV-LS after PLR is not a discriminating criterion to predict fluid responsiveness. Discuss the title in the letter to editor 

Response : Revised title now includes details on the studied population.

Please refer to the letter to editor to find more details about discussion around the title.

2) Methods: fluid responsiveness should be defined in the Methods

Response: We used evaluation of preload dependency to predict fluid responsiveness, but did not assess fluid responsiveness in our cohort as patients did not receive any fluid. In the method section we defined what we considered preload-dependency : a LVOT-VTI variation greater than 10% after a passive leg raising maneuver, a validated method.

3) Results : (page13, line271): This study compared the variation of LV-LS in whole subjets after PLR. The variations of other parameters after PLR such as LV-SSR, RV-LS and LA-LS which were assessed by speckle tracking analysis should be added in this analysis to test their prediction value for fluid responsiveness.

Response: These data are now presented in the results section.

4) Results(page13, line270-274): Significant differences of mean LV-LS variations between T0 and T1 was observed in the whole population (n=60, p=0.04), but not in the PLD+ group (n=33, p=0.04) and PLD- group(n=27, p=0.25). Additionally, these variations of LV-LS after PLR were not statistically different between the groups (p=0.08)? The results might result from the small numbers of subjects, and statistical power test should be added in this study.

Response: We calculated the number of required subjects according to our primary goal which was to determine the sensitivity and specificity of LV-LS variations to assess preload dependency. We assumed, in accordance with literature, that 50% of our ACF population would be PLD+ and 50% PLD-, with roughly balanced groups. Taking that into account, for an aera under the curve > 0.85 with a confidence interval of 0.20 (CI 95% 0.75-0.95) and an alpha risk of 5% we found a required number of subjects of 60 with 50% in each group. Required number of subjects could not be calculated for the LV-LS variation, as we had no idea if 1) it would vary , and 2) in what proportion. These variations were never studied before. The “a posteriori” calculated required number of subjects, that would have been needed to ascertain a difference in LV-LS variation between our groups is 400. We, for sure, must acknowledge a lack of power... We added this important limitation in the discussion section. 

5) Is there any liner correlation between variations of LV RVOT VTI and LV-LS after PLR?

Response : Unfortunatelly, no statistical correlation was found between LVOT-VTI variations and LV-LS variations after PLR.

- In the whole population (n=60) the Pearsons’ coefficient was 0.183, p = 0.161.

- In the PLD+ group (n=33) the

Pearsons’ coefficient was -0.015, p = 0.934

- In the PLD- group (n=27) the

Pearsons’ coefficient was -0.051, p = 0.800 

6) Conclusion(page18, line384-385): This study did not test the correlations between LV-LS and afterload and intrinsic myocardial function. You should add related statistical analysis or modify the conclusion.

Response: Noted, we modified the conclusion

7) What is the advantages and innovations of this study?

Response: We built this observational study after the results of Nafati and colleagues’ study was published [21].

Indeed, that was the first time a study showed a variation of LV-GLS after a volume expansion in a preload dependent population of patients. We decided to question the problem from the other side and evaluate LV-LS variation after PLR as a predicting marker of preload dependency.

In current practice, LVOT-VTI variation after PRL is already a reliable marker, easy to learn and available at bedside in most situations. From that point of view, evaluation of the LV-LS variation could appear pointless.

But, two points must be discussed here.

First, this study shows that the Speckle tracking technology can be used in ICUs in clinical practice when available, as a simple tool that can precisely evaluate the systolic segmental function of the myocardium, following myocardial strain after severe myocarditis for example. 

Cardiologists now use it as a routine evaluation.

Second, our study raises a caution here : interpretation of myocardial strain parameters in ICU patients implies that clinician should assess the haemodynamic status of the patients before analyzing any myocardial strain measurements. 

Other : Journal requirements 

PLOS ONE's style requirements: Instructions followed 

What type of consent you obtained : Written consent was obtained after oral and written information

Competing interest section: Completed. Marc Leone, co-author, added his competing interest in this section as well.

---

## [Decision Letter · Decision Letter 1]

4 Aug 2021

PONE-D-21-11995R1

Left ventricular longitudinal strain variations assessed by speckle-tracking echocardiography after a passive leg raising maneuver in patients with acute circulatory failure  to predict fluid responsiveness: A prospective, observational study

PLOS ONE

Dear Dr. ROY,

Thank you for submitting your manuscript to PLOS ONE. After careful consideration, we feel that it has merit but does not fully meet PLOS ONE’s publication criteria as it currently stands. Therefore, we invite you to submit a revised version of the manuscript that addresses the points raised during the review process.

Thank you for your revisions. I am in agreement with Reviewer #1 in that the paper would be strengthened by including in the manuscript many of the responses to the reviewer comments the explain the underlying reasoning for many of the decisions that were made.

We look forward to receiving your revised manuscript.

Kind regards,

Robert Ehrman, MD, MS

Academic Editor

PLOS ONE

Journal Requirements:

Additional Editor Comments (if provided):

Reviewers' comments:

Reviewer's Responses to Questions

**Comments to the Author**

1. If the authors have adequately addressed your comments raised in a previous round of review and you feel that this manuscript is now acceptable for publication, you may indicate that here to bypass the “Comments to the Author” section, enter your conflict of interest statement in the “Confidential to Editor” section, and submit your "Accept" recommendation.

Reviewer #1: (No Response)

Reviewer #2: (No Response)

2. Is the manuscript technically sound, and do the data support the conclusions?

Reviewer #1: Yes

Reviewer #2: Yes

3. Has the statistical analysis been performed appropriately and rigorously? 

Reviewer #1: Yes

Reviewer #2: No

4. Have the authors made all data underlying the findings in their manuscript fully available?

Reviewer #1: Yes

Reviewer #2: Yes

5. Is the manuscript presented in an intelligible fashion and written in standard English?

Reviewer #1: Yes

Reviewer #2: Yes

6. Review Comments to the Author

Reviewer #1: I thank the authors for responding to the comments raised by the editor and reviewers. I agree with many of their responses. However, the reason for raising these comments is that I expect other interested readers may ask similar questions. Therefore, while responding to the comments satisfies me, other readers won't have the same information unless they delve into the review history to see this. Please consider some minor revisions to the manuscript to include the important information you provide in your response.

Most of my comments were addressed obliquely with revisions made for the editor or other reviewer--there is mention of possible selection bias, and mention of the accuracy of PLR. However, the discordance between E/e' and PLR is of interest, and the authors respond wonderfully with a succinct description of the physiologic basis for why this could be discordant. I'd like to see that in the discussion. It would strengthen the plausibility of the findings and educate the readers. This merges well with the discussion of patient-level heterogeity.

Reviewer #2: (No Response)

7. PLOS authors have the option to publish the peer review history of their article (what does this mean?). If published, this will include your full peer review and any attached files.

Reviewer #1: No

Reviewer #2: No

---

## [Author Response · Author response to Decision Letter 1]

6 Sep 2021

*Academic Editor's remark:

Thank you for your revisions. I am in agreement with Reviewer #1 in that the paper would be strengthened by including in the manuscript many of the responses to the reviewer comments the explain the underlying reasoning for many of the decisions that were made.

*Reviewer#1 : I thank the authors for responding to the comments raised by the editor and reviewers. I agree with many of their responses. However, the reason for raising these comments is that I expect other interested readers may ask similar questions. Therefore, while responding to the comments satisfies me, other readers won't have the same information unless they delve into the review history to see this. Please consider some minor revisions to the manuscript to include the important information you provide in your response.

Most of my comments were addressed obliquely with revisions made for the editor or other reviewer--there is mention of possible selection bias, and mention of the accuracy of PLR. However, the discordance between E/e' and PLR is of interest, and the authors respond wonderfully with a succinct description of the physiologic basis for why this could be discordant. I'd like to see that in the discussion. It would strengthen the plausibility of the findings and educate the readers. This merges well with the discussion of patient-level heterogeneity.

** Answer and amendments :

Thank you for these perceptive remarks that helped us to clarify the discussion. In this new version of the manuscript, we tried to give more details to the readers about how the echocardiographic markers evaluated in ICU patients with acute circulatory failure, should be interpreted with caution. 

Indeed, most markers that we use to assess preload-dependency, left ventricular end diastolic pressure, systolic or diastolic ventricular function, should be interpreted within the framework of the clinical history of the patient, and especially the aetiology of the circulatory failure. 

Thus, we added a paragraph in the discussion section responding to reviewer#1 sensible remark concerning the relationship between E/e’ and preload, and about the E/e’ variations after a passive leg raising maneuver.

We also insisted, on the importance of the patient-level heterogeneity in the presented study, which is probably partly responsible for the negative results.

Therefore, this paragraph was added in the text:

“Regarding the usual echocardiographic markers that were evaluated at T0 and T1, there was a significant difference concerning cardiac output (and surrogates as LVOT-VTI) between the PLD+ and PLD-, but no significant difference was observed concerning echocardiographic markers of left-ventricular end diastolic pressure (LVEDP). Indeed, there was no difference between mean E, E’ lat, nor E/E’lat between PLD+ and PLD- and no significant variation of these markers between T0 and T1 in the whole population. Yet, some authors showed a relationship between E’lat and preload. Mahjoub et al. [45] showed a significant increase of E’lat (and a resultant decrease in E/E’lat) in a population of septic shock patients with diastolic dysfunction (E’Lat < 0.12cm/s) receiving a 500 ml crystalloid administration. These results suggest an improvement of LV relaxation with the correction of hypovolemia in patients with septic shock and diastolic dysfunction. Lamia et al. [14] studied the variations of echocardiographic markers of preload and LVEDP using a PLR maneuver, followed by a volume expansion and found similar results as ours, i.e. a non-significant variation of E, E’Lat and E/E’lat between responders and non-responders following PLR or crystalloid infusion. This study also concerned patients with various aetiology responsible for circulatory failure. Thus, if an increase of E’lat can be observed in preload-dependent patients after correcting hypovolemia, E/E’lat variation is not very accurate to quantify small transitory variations of preload and should rather be used as a marker of LVED, to anticipate the risk of fluid overload when a volume expansion is considered to treat ACF.”

---

## [Editor Report · Decision Letter 2]

9 Sep 2021

Left ventricular longitudinal strain variations assessed by speckle-tracking echocardiography after a passive leg raising maneuver in patients with acute circulatory failure  to predict fluid responsiveness: A prospective, observational study

PONE-D-21-11995R2

Dear Dr. ROY,

We’re pleased to inform you that your manuscript has been judged scientifically suitable for publication and will be formally accepted for publication once it meets all outstanding technical requirements.

Kind regards,

Robert Ehrman, MD, MS

Academic Editor

PLOS ONE
---

## [Editor Report · Acceptance letter]

22 Sep 2021

PONE-D-21-11995R2 

Left ventricular longitudinal strain variations assessed by speckle-tracking echocardiography after a passive leg raising maneuver in patients with acute circulatory failure  to predict fluid responsiveness: A prospective, observational study 

Dear Dr. ROY:

I'm pleased to inform you that your manuscript has been deemed suitable for publication in PLOS ONE. Congratulations! Your manuscript is now with our production department. 

Kind regards, 

on behalf of

Dr. Robert Ehrman 

Academic Editor

PLOS ONE